# Invasion Pattern of *Aedes aegypti* in the Native Range of *Ae. albopictus* in Vietnam Revealed by Biogeographic and Population Genetic Analysis

**DOI:** 10.3390/insects13121079

**Published:** 2022-11-23

**Authors:** Cuong Van Duong, Ji Hyoun Kang, Van Vinh Nguyen, Yeon Jae Bae

**Affiliations:** 1Department of Environmental Science and Ecological Engineering, College of Life Sciences, Korea University, Seoul 02841, Republic of Korea; 2Department of Applied Zoology, Faculty of Biology, University of Science, Vietnam National University, Hanoi 100000, Vietnam; 3Korean Entomological Institute, College of Life Sciences, Korea University, Seoul 02841, Republic of Korea

**Keywords:** *Aedes aegypti*, *Aedes albopictus*, invasion, biogeography, genetic diversity

## Abstract

**Simple Summary:**

Over the last few decades, the yellow fever mosquito *Aedes aegypti* has become one of the most abundant and dangerous arbovirus vectors in Asia. In this study, we assessed the invasion pattern of *Aedes aegypti* by investigating its distribution, relative abundance with its closest competitor *Aedes albopictus*, and genetic diversity in Vietnam. Our results revealed that the distribution and abundance of this species are significantly influenced by climate region, larval habitat, and the presence of *Aedes albopictus*. Furthermore, we found a high level of genetic diversity in *Aedes aegypti*, with two major genetic lineages. The results of this study have significant potential for optimizing vector control strategies regarding future mosquito invasion and outbreak prediction in Asia.

**Abstract:**

Since its introduction to Asia, *Aedes aegypti* has coexisted with the native species *Ae. albopictus* and has been reported to transmit several infectious diseases. However, the development of efficient disease prevention and vector control is hindered by the relatively poor understanding of the biogeography and the genetic diversity of *Ae. aegypti* in the region. This study aimed to determine the invasion patterns of *Ae. aegypti* by evaluating the distribution and abundance of *Ae. aegypti* and *Ae. albopictus* in different climatic regions (northern temperate and southern tropical regions) and habitats (domestic, peri-domestic, and natural). We further analyzed the genetic diversity and phylogenetic relationships of *Ae. aegypti* populations in Vietnam using mitochondrial *COI* gene sequences. Both *Aedes* species were observed at most of the study sites, but only *Ae. albopictus* thrived in northern mountainous areas. In sympatric ranges, the individual abundance of the species was influenced by regional climate and habitats. The tropical climate and availability of domestic containers facilitated the dominance of *Ae. aegypti*, whereas temperate climates and natural breeding sites facilitated that of *Ae. albopictus*. In addition, many genetic polymorphisms were detected in the *Ae. aegypti* populations, which formed two distinct genetic groups; however, this genetic diversity is unlikely to be relevant to the invasive success of *Ae. aegypti*. These findings provide insights into the mechanisms and patterns of *Ae. Aegypti* invasion, which depend on the climate and reproductive strategies in the native range of *Ae. albopictus* in Asia.

## 1. Introduction

Recently, interest in researching methods of biological invasions has increased significantly, especially with respect to mosquito species that are capable of transmitting vector-borne diseases [1]. Particularly, *Aedes aegypti* Linnaeus 1762 and *Ae. albopictus* (Skuse 1894), the two most invasive mosquito vectors native to Africa and Southeast Asia, respectively, have attracted increasing attention [2,3]. Interestingly, the progression of urbanization has created sprawling domestic habitats for these mosquitos. Their strong ecological and physiological plasticity have facilitated their current global distribution [4,5,6]. Their invasive spread has serious epidemiological repercussions due to its effects on the transmission of arboviral diseases, such as dengue, chikungunya, and Zika [4,7,8,9,10,11]. However, despite the availability of various preventative measures and treatments, vector control remains the typical and predominant strategy for reducing the incidence of vector-borne diseases [12].

The reliance of native and invasive species on similar ecological niches, as in the case of *Ae. albopictus* and *Ae. aegypti*, results in biological invasion leading to interspecific competition, affecting the distribution and abundance of the resident species. For example, *Ae. albopictus* has shown a competitive advantage over *Ae. aegypti* in the native range of *Ae. aegypti* in Africa [13,14,15] and well-established *Ae. aegypti* habitats in southeastern USA [16] and Brazil [17,18]. A variety of explanations have been proposed for the displacement of *Ae. aegypti* by *Ae. albopictus*, including interspecific competition [1,19,20], mating interference (i.e., satyrization) that benefit *Ae. albopictus* [21], differential adaptability to unfavorable conditions [22], and tolerance to larval parasites [23]. In contrast, several mid-20th century investigations of *Aedes* abundance in southern Asian cities have suggested that *Ae. aegypti* could displace *Ae. albopictus* [24,25]. However, both the underlying mechanisms and potential geographic distribution of such displacements remain unknown. Therefore, a more comprehensive and up-to-date study of the current distributions of the species in different climatic regions and breeding sites in Asia is necessary to determine the influence of macro-geographic environmental variability and habitat preference on the geographic distribution and abundance of the species.

Population genetics significantly contributes to the study of mosquito invasions by providing critical information for tracking invasion routes and putative source populations, measuring and comparing the genetic diversity of native and introduced populations, and investigating the evolution of introduced populations in novel environments [26,27,28]. Additionally, this information could aid in pest management by facilitating the development of effective control programs and improving the efforts to lessen or prevent the introduction and dispersal of harmful mosquitoes, especially *Ae. aegypti*. Thus, a combination of biogeographic distribution and population genetic analyses of invasive species is needed to fully understand the invasion and evolutionary processes of invasive species in novel habitats and regions. Mitochondrial *COI* genes are widely used as molecular markers for population genetic studies in mosquitoes [2,29,30,31,32,33].

In Vietnam, *Ae. aegypti* invasion was first reported in a central province (Hue) in the 1910s [34]. After a century, this species has become the prime mosquito vector responsible for outbreaks of vector-borne diseases such as dengue, chikungunya, and Zika [35,36,37]. Dengue was first reported in 1959 [38] and has subsequently become one of the most widespread mosquito-borne diseases in Vietnam [39,40]. Interestingly, *Ae. albopictus*, which is indigenous to Vietnam, is considered a secondary vector of both dengue and chikungunya [41]. Despite the important role of mosquito species as vectors, the last investigation of the geographic distribution of *Aedes* in Vietnam was conducted over 10 years ago. In addition, the study only sampled used tires, which may not provide a comprehensive evaluation of the relative abundance of the native species or the progress of *Ae. aegypti* invasion because both species utilize water stagnated in various types of containers in urban areas [14,15,38,39,42]. Furthermore, the origin of *Ae. aegypti* populations in Vietnam has not yet been determined by tracing the maternal genetic relationship using the *COI* gene.

The aims of this study were to compare the geographic distribution of the native species *Ae. albopictus* and the invasive species *Ae. aegypti* in Vietnam and to evaluate the genetic diversity of *Ae. aegypti* in Vietnam. The outcomes of this study may provide insights into the colonization process of *Ae. aegypti* in the native range of *Ae. Albopictus*, clarify the roles of regional climate and habitat types in determining the distribution and abundance of *Aedes* species, and elucidate the poorly understood genetic aspects.

## 2. Materials and Methods

### 2.1. Characteristics of Study Area

Vietnam is part of the eastern Indochinese peninsula. The land area extends from 8 to 24° N in latitude, encompassing a great variety of terrain. The South China monsoon also contributes to climatic heterogeneity between the northern and southern areas of the country. In general, Vietnam includes two main climatic regions (northern and southern Vietnam). More specifically, the northern region is characterized by a temperate/subtropical climate with a cold-dry winter, especially in the northern mountainous regions, which often experience subzero temperatures. The southern region is characterized by a tropical climate with two hot seasons (hot-rainy and hot-dry) [43].

### 2.2. Mosquito Sampling and Identification

“Mosquito surveys” were conducted in 18 urban cities distributed along a north–south transect of Vietnam, which were selected based on population density, their associations with *Aedes* mosquitoes, and frequently reported dengue transmission (Figure 1, Table 1) [44,45]. Mosquito specimens were collected in July 2019 and October 2020, corresponding to the rainy season. All containers at the study sites with standing water were inspected for the presence of *Aedes* larvae, and every container that contained at least one mosquito larva was recorded. 

We chose one study site per city such that all study sites had similar habitat conditions and included a variety of potential sampling points, such as residential houses, recreational parks, playgrounds, car repair stations, pagodas, and schools. Each study site was a circular area 5 km in diameter or approximately 78.5 km^2^. We tried to establish 16 sampling points that were spaced equally within the study sites (16 sampling points × 18 study sites = 288 sampling points in total) (Appendix A). The area of each sampling point was approximately 300 m^2^, and all containers with stagnant water were checked and sampled. Although the designed sampling points were often blocked by obstacles, such as buildings or ponds, we tried to maintain a distance of >500 m between the sampling points (Appendix A). Furthermore, to assess the habitat preferences of *Aedes* species at each sampling point, the inspected containers were categorized based on their location and material: domestic (indoor, artificial material containers), peri-domestic (outdoor, artificial material containers), and natural (outdoor, natural-material containers) habitats (Table 2, Figure 2c). The mosquito larvae in each container were swept or pipetted into separate plastic bottles for counting and subsequent genetic analyses. Immature specimens were either reared to maturity in the laboratory (breed in separate cages according to containers and sites, 25–27 °C, 12 h photoperiod) or preserved directly in 80% ethanol. We used a multi-way ANOVA to analyze the abundance of *Ae. aegypti* and *Ae. albopictus* among biogeographic regions and habitat types (Appendix A). Multi-way ANOVA was performed using R version 4.0.5 (https://www.r-project.org/, accessed on 10 October 2021). *Aedes albopictus* samples from several locations were also included in our previous study, which investigated the genetic diversity and population structure of *Ae. albopictus* in Vietnam [46].

### 2.3. Mitochondrial DNA Sequencing and Analysis

The representative study sites that harbored specimens from both species were used to investigate the genetic diversity and population structure of the *Ae. aegypti* populations in both north and south Vietnam. Only one or two specimens from each breeding habitat were included in the genetic analysis to prevent sibling bias; these specimens were preserved in absolute ethanol at −20 °C.

Genomic DNA was extracted from the legs of adult specimens and abdominal muscles of larval specimens using the DNAeasy Blood & Tissue kit (QIAgen, Hilden, Germany), by following the manufacturer’s recommendations. A 658-bp fragment of the mtDNA *COI* gene was PCR-amplified from each DNA sample using the AccuPower PCR PreMix kit (Bioneer Co., Daejeon, Republic of Korea) and a previously published universal primer pair (LCO1490: 5′-GGTCAACAAATCATAAAGATATTGG-3′; HCO2198: 5′-TAAACTTCAGGGTGACCAAAAAATCA-3′) [47]. The amplification conditions were: 95 °C for 2 min, followed by 35 cycles of 95 °C for 30 s, 50 °C for 30 s, and 72 °C for 30 s, and a final extension step at 72 °C for 2 min. Amplification was confirmed by performing gel electrophoresis using 1.5% agarose gels in TAE 1X buffer and visualizing under UV light. The validated PCR products were then purified enzymatically using Exonuclease I to degrade the residual PCR primers and shrimp alkaline phosphatase to dephosphorylate the remaining dNTP (New England BioLabs, Ipswich, MA, USA). The purified product was bidirectionally sequenced using an ABI Prism Sequencer 3130 (Applied Biosystems, Foster City, CA, USA). The resulting sequences were manually corrected to ensure correct alignment along codons in both reverse and forward directions using MEGA v.7 [48]. 

The number of haplotypes, number of polymorphic sites (S), haplotype diversity (Hd), and nucleotide diversity (π) were calculated using DnaSP v. 6.12.03 [49]. Tajima’s *D* and Fu’s *Fs* were estimated for each population using ARLEQUIN v. 3.5.2.2 to establish non-neutral evolution and deviation from mutation-drift equilibrium, with population expansion signatures inferred using significance values that were calculated by generating 1000 random samples [50]. To investigate the demographic history, pairwise mismatch distributions were inferred using a population growth-decline model in DnaSP. Genetic structure was estimated through hierarchical analysis of molecular variance (AMOVA) in ARLEQUIN with 1000 permutations and populations grouped by geographic regions as defined in Table 3. This included (1) three groups (North, Center, and South) and (2) two groups (North and Center + South) (Appendix A). We assessed statistical significance based on the degree of differentiation among regions (Φ_CT_), among populations within regions (Φ_SC_), and within populations (Φ_ST_) using permutation tests with 1000 random permutations.

A haplotype network was inferred using the nexus file from DnaSP and PopArt v. 1.7. software (http://popart.otago.ac.nz/index.shtml, accessed on 23 September 2021) [51].

Furthermore, to track the invasion pathways of *Ae. aegypti* in Vietnam, the phylogenetic relationship between haplotypes from Vietnam and those from native and invasive *Ae. aegypti* populations worldwide (http://blast.ncbi.nlm.nih.gov/, accessed on 23 September 2021; Appendix A) were evaluated. A maximum-likelihood (ML) phylogenetic tree was constructed using MEGA v.7 with 1000 bootstrap replicates. The best-fit nucleotide substitution model was inferred from three criteria (Akaike information criterion, corrected Akaike information criterion, and Bayesian information criterion) using jModelTest v2.1.10 [52]. The HKY+I model was selected as the most suitable model. 

## 3. Results

### 3.1. Habitat, Abundance, and Distribution of Aedes Species in Vietnam

A total of 848 available containers (44.67 ± 4.8 containers per study site) with standing water were identified in the 18 study sites along a north–south transect in Vietnam (Table 1 and Table 2); most containers (n = 690, 81.4%) were found to harbor immature *Aedes*. Peri-domestic habitats accounted for the majority (n = 450, 65.2%) of the study sites, followed by natural (n = 122, 17.7%) and domestic habitats (n = 118, 17.1%; Table 2). Used tires were the most frequently sampled source of standing water in peri-domestic habitats, as were flowerpots and ceramic jars in domestic habitats, and bamboo nodes in natural habitats (Table 2). 

*Aedes albopictus* and *Ae. aegypti* accounted for 58.7 and 41.3%, respectively, of the 17,178 immature specimens collected during this study (Table 1). Overall, they were most frequently found in peri-domestic containers (78.5%), followed by standing water sources in the domestic and natural habitats (16.8 and 4.7%, respectively). In each breeding type, the relative abundance varied significantly. In particular, *Ae. aegypti* were found more frequently than was *Ae. albopictus* in domestic containers (90 and 57 containers, respectively). In contrast, peri-domestic containers had a higher frequency of 68.9% (310 containers) for *Ae. albopictus* compared with 31.1% (153 containers) for *Ae. aegypti*. Similar results were observed for natural breeding types, with 106 containers containing *Ae. albopictus* larvae, whereas the *Ae. aegypti* larvae were found in 12 containers (Table 2). 

*Aedes albopictus* was observed at all 18 study sites, indicating its ubiquitous distribution in Vietnam, whereas *Ae. aegypti* was mainly distributed in southern Vietnam and was absent from study sites S1–S6 in the northern mountainous regions (Figure 2a, Table 1). Furthermore, statistically significant differences were observed between the frequencies of *Ae. albopictus* and *Ae. aegypti* among climatic regions and habitats or both factors (multi-way ANOVA, *p* < 0.05; Figure 3, Appendix A). Specifically, the presence and abundance of *Ae. aegypti* were negatively affected by the presence and abundance of *Ae. albopictus* in certain climatic regions and breeding habitats (AMOVA test, *p* < 0.00001; Figure 3 and Appendix A). For example, *Ae. aegypti* were dominant over *Ae. albopictus* at the southern sites (tropical region, S12–S18; 868 ± 276.5 vs. 401 ± 131.1 individuals/site; Figure 2), whereas *Ae. albopictus* was dominant over *Ae. aegypti* in the northern sites (720.7 ± 290.9 vs. 202.8 ± 156.5 individuals/site). Furthermore, *Ae. aegypti* were dominant over *Ae. albopictus* in almost all domestic container types at sites where both species coexisted (S7–S18) and in peri-domestic containers at the southern sites (S12–S18; Figure 2b). In contrast, *Ae. albopictus* was dominant in all natural standing water sources, regardless of site, and in peri-domestic standing water sources at the northern sites.

Other species, including *Armigeres subalbatus* (Coquillett, 1898), *Culex quinquefasciatus* Say, 1823, *Toxorhynchites splendens* (Weidemann, 1819), *Tripteroides aranoides* (Theobald, 1901), and *Aedes annandalei* (Theobald, 1910), were also observed in *Aedes* habitats, but were observed to occur only in a few containers at very low densities, suggesting the dominance of *Aedes* species in such breeding habitats.

### 3.2. Population Genetics and Phylogenetic Relationships of Aedes aegypti in Vietnam

Analysis of the *COI* sequences from 122 *Ae. aegypti* specimens yielded 29 haplotypes, which were distinguished by 34 variable sites (4.7%); of these, 25 were parsimony-informative (73.5%; Table 3). Seven to twelve haplotypes were detected in each local population, with the highest number of haplotypes (n = 12) detected in S17 and the lowest number (n = 7) detected in sites S9, S12, and S15. The overall Hd and π of *Ae. aegypti* were 0.873 and 0.0095, respectively. Hd ranged from 0.7905 in S08 to 0.9412 in S17, whereas *π* ranged from 0.0074 in S12 to 0.0112 in S7. Neutrality tests (Tajima’s *D* and Fu’s *Fs*) yielded insignificant results (*p* > 0.05) for the population as a whole or each local population individually, indicating that the observed polymorphisms were consistent with the neutral mutation model (Table 3, Figure 4).

The AMOVA indicated that most of the total genetic variance could be attributed to within-population variation when all populations were grouped into three regions, thereby suggesting a very weak genetic structure and high gene flow among the *Ae. aegypti* population in Vietnam (Φ_ST_ = 99.36, *p* = 0.43; Appendix A). However, when all populations were divided into two geographic clusters (i.e., north and south), the AMOVA analysis indicated significant spatial genetic structuring (Φ_CT_ = 4.42, *p* < 0.05); however, most of the variance still occurred within the populations (Φ_ST_ = 97.88, *p* = 0.43; Appendix A).

The haplotype network inferred for *Ae. aegypti* populations indicated two main genetic groups: I and II. Group II was further separated into two subgroups: IIa and IIb (Figure 5a). Nucleotide divergence ranged from 0.010 (between Subgroups IIa and IIb) to 0.022 (between Group I and Subgroup IIa). The divergence between Groups I and Subgroup IIb was 0.018. Group I was the most diverse with 17 haplotypes, of which three haplotypes (H1, H2, and H7) were dominant. H1 was the most common haplotype, constituting 29.5% of the haplotypes in the 122 specimens analyzed and was detected at all 18 study sites (Figure 5a). H1 has also been detected in invasive populations around the world, including in America (Canada, Mexico, Peru, Puerto Rico, and Chile), Asia (Pakistan, India, Malaysia, Sri Lanka, and Cambodia), Europe (United Kingdom, Germany, and Portugal), and Oceania (New Caledonia) (Appendix A). Meanwhile, H2 was the second most common haplotype, constituting 14.8% of the haplotypes in the analyzed specimens; H2 has also been detected in Australia. Haplotype H7 has been detected in both Pakistan and Australia (Appendix A).

Subgroup IIa included seven haplotypes that were widespread among the study sites. Haplotype H3 was the most common one and has been detected in Ecuador and Mexico (Appendix A). Subgroup IIb was the smallest group comprising three haplotypes that have not been previously reported and are restricted to the northern sites (S7, S8, and S9) (Figure 5a). 

An ML phylogenetic tree, which was constructed to evaluate the broader relationships between the haplotypes detected in Vietnam and those previously reported in other populations worldwide (Appendix A), also indicated two well-supported mitochondrial groups (I and II); the second group (Group II) was divided into two subgroups (IIa and IIb; Figure 5b). Interestingly, the haplotypes from Vietnam were more closely associated with haplotypes from other invasive populations rather than with those of the native range of the species in Africa. In particular, Group I was closely related to haplotypes from invasive populations in Asia (India, Pakistan, Thailand, and Cambodia) and America (Venezuela, Panama, and USA). Subgroup IIa was closely related to haplotypes from America (Mexico, Martinique, and Brazil) and Asia (Thailand). Subgroup IIb was related to haplotypes from native *Ae. aegypti* populations in Kenya and those of invasive populations in Panama (Figure 5). 

## 4. Discussion

The replacement of *Ae. aegypti* by *Ae. albopictus* has recently been attributed to the superior competitive ability of *Ae. albopictus* larvae [1,19,22] and the negative effects of interspecific mating on *Ae. aegypti* fitness [21]. In contrast, this study clearly demonstrates that *Ae. aegypti* is a superior competitor in Asia based on the abundance of the species in Vietnam. This phenomenon has also been reported in other studies conducted in southern Asia [24,25]. However, the mechanisms underlying this pattern are yet to be fully elucidated. Interspecific resource competition by sharing breeding habitats can be excluded because only 26 of the 664 containers contained larvae of both *Aedes* species (Table 2). Therefore, we infer that local climate or adult habitat preference may affect distribution and relative abundance of the *Aedes* species. In terms of the effects of climate, the invasive species *Ae. aegypti* were predominant in the southern (tropical) sites, whereas the native species *Ae. albopictus* accounted for a larger proportion of immature mosquitoes observed at the northern (i.e., temperate) sites (Figure 2a). The interaction between species and regional climate was statistically significant (*p* < 0.0001; Appendix A), suggesting that the displacement of *Ae. albopictus* by *Ae. aegypti* is regionally dependent, at least in Vietnam. This pattern may be due to climatic differences between northern and southern Vietnam, which results in different species competition outcomes. For example, the year-round hot weather in southern Vietnam, which is characterized by hot dry and hot rainy seasons, may favor the successful establishment of *Ae. aegypti*, which are more tolerant to dry and hot conditions than the *Ae. albopictus* [32,53,54]. Meanwhile, the climate of northern Vietnam, which experiences four seasons including cold winters, may limit the development of *Ae. aegypti*, which is sluggish at temperatures below 17 °C [55]. In the northern mountainous regions of Vietnam (site S1–S6 in Figure 2a), high elevation and influences of the northeast winter monsoon result in considerable periods of long and cold cloudy conditions [43,56]. The survival of such conditions necessitates photoperiod diapause ability, which *Ae. aegypti* are incapable of [57]. Consequently, these climatic conditions may influence the invasion success and, more specifically, prevent *Ae. aegypti* from invading northern mountainous Vietnam. The current distribution of these two *Aedes* species in Vietnam is similar to that reported in 2010 [44], which suggests that the climate of northern Vietnam and/or the competitive superiority of *Ae. albopictus* played a crucial role in preventing northward expansion by *Ae. aegypti*. However, the geographic distribution of *Ae. aegypti* in Vietnam should be continuously monitored, because increasing urbanization and climate change could facilitate future invasions [58]. 

The preference of breeding habitat could also affect the relative abundance of *Aedes* species. *Ae. aegypti* generally prefers indoor containers, whereas *Ae. albopictus* prefers outdoor water sources near vegetated areas in distributional ranges [18,55]. Indeed, this study showed the habitat-dependent displacement of *Ae. albopictus* by *Ae. aegypti* in Vietnam. In particular, *Ae. aegypti* was dominant in peri-domestic water sources in southern Vietnam and in most domestic water sources in areas where both species occurred, whereas *Ae. albopictus* maintained its dominance in natural water sources and peri-domestic habitats of northern Vietnam (Figure 2). A multi-way ANOVA also revealed a significant interaction between species and habitat (*p* < 0.0001; Figure 3, Appendix A), which suggests that *Ae. aegypti* remains mainly associated with artificial containers and has not yet adapted to natural habitats in Vietnam. However, observations of *Ae. aegypti* in several natural breeding habitats in southern Vietnam indicate that this species has the potential to invade a wider range of habitats under favorable conditions (i.e., tropical climates). Furthermore, when alien species encounter environments that differ from their native ranges, they need to adapt to local environments [59]. The findings of this study agree with the hypothesis of pre-adapted behavioral traits of domestic *Ae. aegypti* that enable it to be a successful invader, such as: (1) house entry (endophily), (2) preference for oviposition in human-made domestic containers, and (3) preference for blood-feeding from humans [20]. Indeed, in northern Vietnam, under unfavorable conditions, *Ae. aegypti* were dominant in domestic containers in locations where close contact with humans was more likely to occur (Figure 2b, Table 2). This behavior could be a hidden strategy that results in reduced interspecific competition in mating interference, blood sources, larval competition, or other factors (e.g., blood sources and temperature). Moreover, domestic habitats could also function as footholds from which *Ae. aegypti* could expand further under favorable environmental conditions (i.e., summer). Among the domestic containers, flowerpots and ceramic jars were the most frequent oviposition containers for *Ae. aegypti*. Meanwhile, both *Aedes* species utilized used tires regardless of the region. This habitat information could be important for *Aedes* surveillance and its control in urban areas in Asian countries, especially in Vietnam. Additional laboratory and field investigations focusing on the interactions of the *Aedes* species and effects of environmental factors (e.g., temperature, precipitation, seasonal changes, food resources, mating, and satyrization) on the outcomes of their competition are needed to elucidate the mechanisms underlying species displacement and facilitate the prediction of future range expansion of *Aedes* in Asia. Furthermore, because this study was conducted in urban areas across Vietnam, investigations must be undertaken in other areas, such as rural or forest areas, to examine the invasion pattern of *Ae. aegypti*, especially in the southern ranges.

The genetic characteristics of the invasive populations strongly influence the success of invasion. Accordingly, one of the aims of this study was to investigate the invasion patterns of *Ae. aegypti* using genetic analyses. Interestingly, multiple genetic lineages were detected (Figure 4a,b). *Ae. aegypti* populations are historically diverged and admixed within their native ranges, owing to a history of environmental change [31]. Therefore, the *Ae. aegypti* population in Vietnam may inherit ancestral variation either through a large invading gene pool or multiple introductions. Furthermore, the demographic analysis yielded insignificant Tajimas’ *D* and Fu’ *Fs* test results (Table 3) and a biphasic mismatch distribution that probably corresponded to two successive population size increases (Figure 4). These findings support the rejection of the recent expansion hypothesis and indicate multiple introductive events or a large invading genetic pool. Moreover, AMOVA revealed very weak geographic structuring among the northern and southern populations, suggesting that individuals frequently transfer between populations (Appendix A). This finding, however, contrasts with that of a previous study that investigated the genetic structure of *Ae. aegypti* in Vietnam and failed to detect any differentiation among populations [60]. This discrepancy could be the result of the restriction of Subgroup IIb to a few individuals in the northern regions (Figure 5c) that may have been locally adapted or recently introduced. Evidence exists for a possible adaptive advantage for *Ae. aegypti* populations in heterogeneous environments [61]; therefore, the specific prevalence of Subgroup IIb in temperate/subtropical regions of Vietnam should be further investigated to evaluate the local adaptation of this and other genetic lineages with regard to expansion capability, especially into northern Asia and other temperate regions. 

Previous studies have addressed the positive effects of genetic diversity on the population size of species [62]. The outcomes of this study, along with those of a previous study on the genetic diversity of *Ae. albopictus* in Vietnam [46], partly support this hypothesis. In particular, despite the genetic diversity of *Ae. aegypti* at the northern sites (S7, S8, and S9) being similar to that at the southern sites (S12, S14, S15, S17, and S18), its density was significantly lower in the northern sites (Appendix A). Therefore, interspecific competition or local adaptation, rather than the level of genetic diversity, could be responsible for the lower abundance of *Ae. aegypti* larvae in northern Vietnam. In contrast, the population size of *Ae. albopictus* in Vietnam is possibly affected by both the genetic diversity and invasion of *Ae. aegypti*. In fact, the abundance of *Ae. albopictus* was highest in the regions that also had the highest genetic diversity (study sites S7–S11 in Figure 1, Appendix A) and lower in regions with lower genetic diversity (Appendix A). In addition, tropical regions are favorable for *Ae. albopictus* development [63]. The surveys conducted in this study found that *Ae. albopictus* abundance was the least in southern Vietnam (Appendix A), which suggests that the abundance of this species was negatively affected by the high prevalence of *Ae. aegypti*.

This study used the phylogenetic analysis of *COI* sequences to identify the invasion sources for *Ae. aegypti* in Vietnam. The analysis provided evidence for many different dispersal pathways in Vietnam, but mostly from invasive populations rather than native populations. The major haplotypes (H1, H2, H3, and H7), which accounted for 60.7% of all specimens analyzed in this study, have been reported in many invasive populations in the Americas, Asia, and Europe. Therefore, these major haplotypes are highly adaptive to novel environments and could be primary invasion sources, not just in Vietnam, but worldwide. Moreover, the phylogenetic tree revealed two deeply divergent groups (I and II), with Group II further divided into subgroups IIa and IIb (Figure 5). The presence of these two groups in Vietnam could be the result of multiple introductions or a single introductive event with a large genetic propagule. The findings of this study indicate that genetic diversity varied between groups, with Group I being the most diverse (17 haplotypes), followed by Subgroup IIa (six haplotypes) and Subgroup IIb (three haplotypes) (Figure 5). This suggests the long-term introduction and establishment of haplotypes from Group I rather than those of Group II in Vietnam; Subgroup IIb seems to be the newest arrival. Therefore, the colonization of Vietnam by *Ae. aegypti* is likely ongoing, and different genetic lineages have invaded the country, perhaps at different times and possibly through different routes. The phylogenetic tree constructed in this study was unable to strongly support the origin of *Ae. aegypti* lineages in Vietnam; however, most of the detected haplotype groups have also been associated with invasion in other parts of the world. Taken together with the major haplotype similarity results, the *Ae. aegypti* population in Vietnam is likely to have been introduced through other invasive populations. Future studies using high-resolution multilocus genomic data, a larger scale of sampling, and sampling in both native and invasive regions are needed to elucidate the invasion history of *Ae. aegypti* in Vietnam. Such studies could help determine whether genetic clusters were introduced in a series of invasions or in a single event.

## 5. Conclusions

By combining biogeographic and population genetic approaches, this study provides novel insights into the introduction patterns, expansion, evolutionary history, and genetic structure of the invasive mosquito *Ae. aegypti* in the native range of its closest competitor *Ae. albopictus* in Vietnam. This study highlights the success of *Ae. aegypti* invasion in a part of Asia that has significant variation in local climate and habitat preference. Furthermore, the contrasting distribution and abundance patterns of the two *Aedes* species indicate an ongoing displacement process in the region. The sources of *Ae. aegypti* in Vietnam are likely to be genetically similar to other invasive *Ae. aegypti* species worldwide. Furthermore, we observed a high level of genetic diversity throughout the country. However, the level of genetic diversity is not likely to be responsible for the invasive success of this species.

## Figures and Tables

**Figure 1 insects-13-01079-f001:**
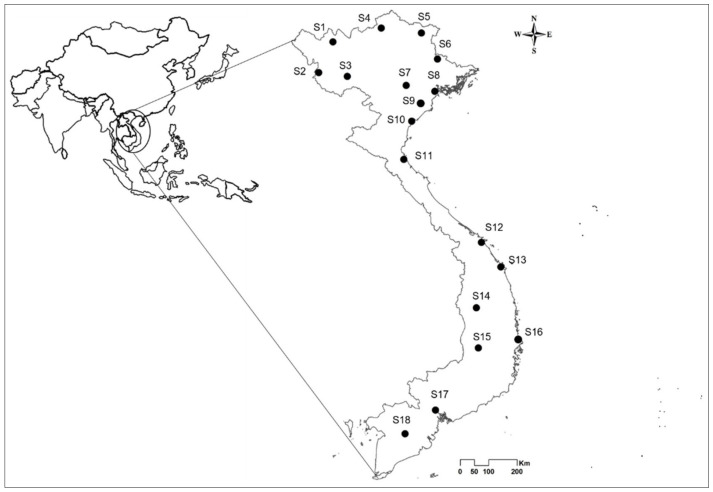
Map of study sites for *Ae. aegypti* and *Ae. albopictus* across Vietnam. Site codes are defined in Table 1.

**Figure 2 insects-13-01079-f002:**
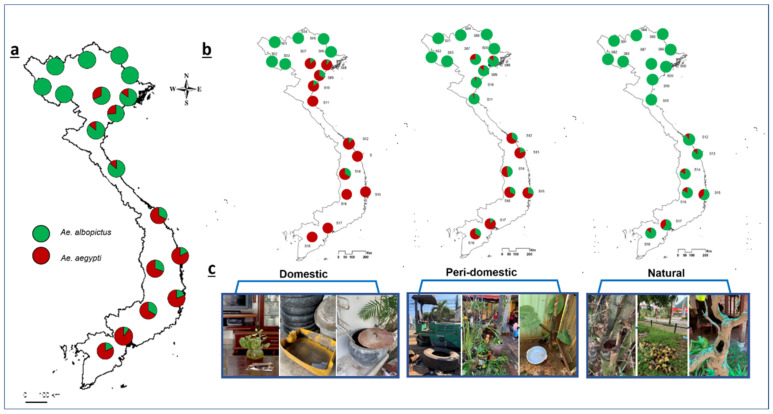
Relative abundance of *Aedes* species within study sites in Vietnam. (**a**) Relative abundance of *Ae. aegypti* and *Ae. albopictus* across all habitat types. (**b**) Relative abundance of *Ae. aegypti* and *Ae. albopictus* within specific habitat types (domestic, peri-domestic, and natural). (**c**) Examples of containers found in each habitat type.

**Figure 3 insects-13-01079-f003:**
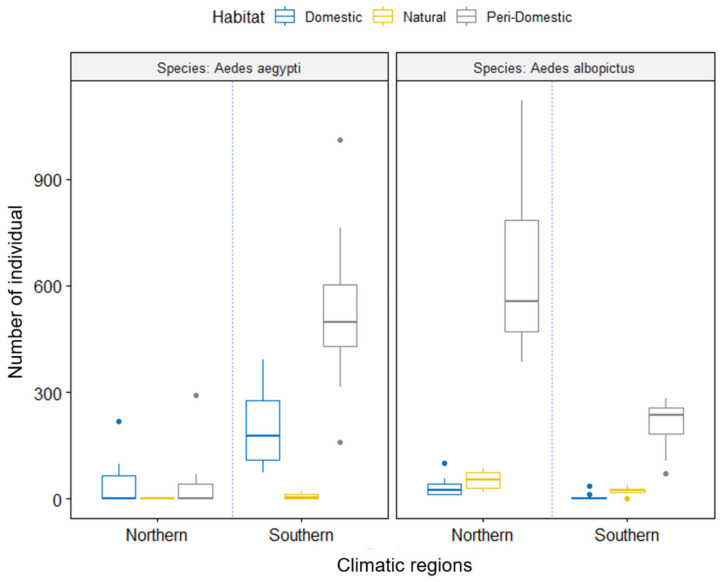
Effects of habitat and region on the abundance of *Ae. aegypti* and *Ae. albopictus* (multi-way ANOVA). Color circles are indicated for outlier data.

**Figure 4 insects-13-01079-f004:**
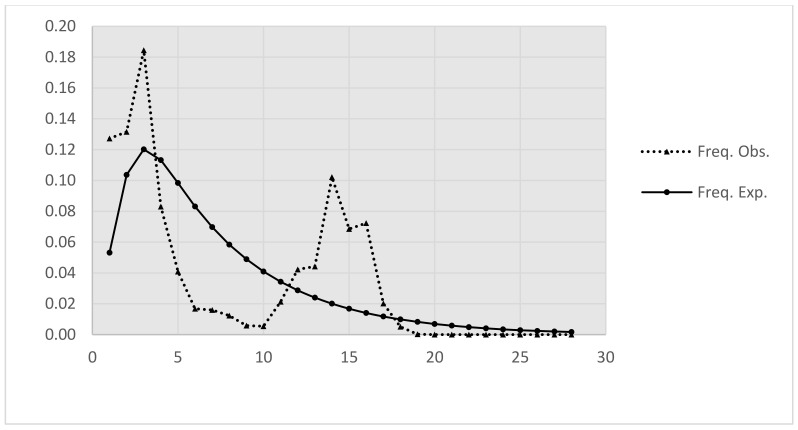
Mismatch distribution of *Aedes aegypti* populations in Vietnam. The dotted line represents the observed distribution of pairwise differences between pairs of mitochondrial *COI* haplotypes, whereas the solid line illustrates the expected distribution under the population expansion.

**Figure 5 insects-13-01079-f005:**
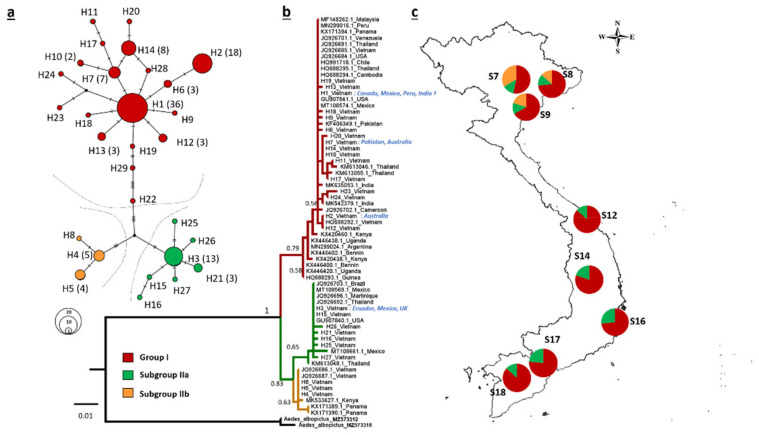
Population genetics of *Aedes aegypti* in Vietnam. (**a**) Network of *COI* haplotypes detected in *Ae. aegypti* specimens (n = 122) collected in Vietnam. Values in parentheses indicate frequencies > 1, and black dots represent inferred intermediate haplotypes. (**b**) Maximum-likelihood phylogenetic tree of *COI* haplotypes. Branch values indicate bootstrap values. (**c**) Distribution map of the *COI* haplotype groups.

**Table 1 insects-13-01079-t001:** Distribution of *Aedes aegypti* and *Aedes albopictus* specimens among 18 study sites in Vietnam.

ID	Localities	Latitude	Longitude	Collection Date	Number of Individuals	
					*Ae. aegypti* (%)	*Ae. albopictus* (%)
S1	Lai Chau	22.38825	103.46001	July 2019	0 (NC)	1431
S2	Dien Bien	21.42103	103.00957	July 2019	0 (NC)	859
S3	Son La	21.29897	103.91367	July 2019	0 (NC)	1025
S4	Ha Giang	22.82850	104.98580	July 2019	0 (NC)	711
S5	Cao Bang	22.670478	106.255141	July 2019	0 (NC)	626
S6	Lang Son	21.84563	106.76363	July 2019	0 (NC)	617
S7	Hanoi	20.991845	105.802818	June 2019	509 (34.9)	951 (65.1)
S8	Hai Phong	20.79475	106.98955	June 2019	169 (27.9)	437 (72.1)
S9	Nam Dinh	20.410715	106.159643	October 2020	153 (22.3)	532 (77.7)
S10	Thanh Hoa	19.88138	105.95152	October 2020	103 (15.4)	565 (84.6)
S11	Nghe An	18.676604	105.70048	October 2020	80 (13.7)	504 (86.3)
S12	Da Nang	16.05247	108.15341	October 2020	633 (66.8)	315 (33.2)
S13	Quy Nhon	13.771584	109.224653	October 2020	1144 (78.9)	305 (21.1)
S14	Nha Trang	12.242141	109.185087	October 2020	967 (79.5)	249 (20.5)
S15	Gia Lai	13.990133	107.992312	October 2020	591 (65.8)	307 (34.2)
S16	Dak Lak	12.67239	108.042828	October 2020	555 (66.2)	284 (33.8)
S17	Sai Gon	10.754719	106.701183	October 2020	1016 (88.5)	132 (11.5)
S18	Can Tho	10.005611	105.741194	October 2020	856 (76.7)	260 (23.3)

ID, locality code used for the downstream analyses; NC, not computed.

**Table 2 insects-13-01079-t002:** Distribution of *Aedes aegypti* and *Aedes albopictus* specimens among larval habitats and standing water sources at 18 study sites in Vietnam.

Type of Containers		Number of Positive Containers per Species and Number of Individuals			N_P_/N_T_ (%)	Total No. of Individual
		*Ae. albopictus* (N)	*Ae. aegypti* (N)	Shared (N)		
Domestic	Flowerpot/vase	9 (125)	43 (997)		52/67 (77.6)	1122
	Ceramic jar	8 (284)	33 (135)		41/45 (91.1)	1414
	Plastic containers (bucket, pot, trash, …)	15 (196)	14 (185)		29/35 (82.9)	381
Peri-Domestic	Used tires	144 (3677)	94 (3321)	21	238/284 (83.8)	6998
	Cement tanks	15 (246)	1 (11)		16/20 (80.0)	257
	Plastic containers (bucket, pot, …)	66 (1171)	30 (693)	4	96/131 (73.3)	1864
	Bonsai tank	48 (2752)	4 (140)	1	52/78 (66.7)	2892
	Polythene sheet	11 (142)	0 (NC)		11/11 (100)	142
	Pagoda/garden ornament	4 (113)	7 (488)		11/12 (91.7)	601
	Barrel	8 (554)	1 (28)		9/9 (100)	582
	Others (glass vials, orchid basket, styrofoam box, etc.)	13 (181)	4 (37)		17/26 (65)	218
Natural	Rock holes	6 (35)	0 (NC)		6/7 (85	35
	Tree holes	14 (47)	3 (7)		17/19 (89.5)	54
	Leaf axils	3 (31)	0 (NC)		3/4 (75)	31
	Bamboo joints	53 (442)	6 (32)		59/62 (95.2)	474
	Coconut shells	26 (145)	3 (22)		29/34 (85.3)	167
	Ground pools	4 (46)	0 (NC)		4/4 (100)	46
Total		447 (10,182)	243 (7,096)	26	690/848 (81.4)	17,278

N, number of individuals; N_P_, number of positive containers; N_T_, total number of inspected containers; NC, not computed.

**Table 3 insects-13-01079-t003:** Genetic diversity indices of *Aedes aegypti* from selected study sites in Vietnam, based on *COI* sequences (n = 122).

ID	Region	Locality	Life-Stages Analyzed	N	No. Halotypes (N_H_)	Halotype Diversity (*Hd*) (SD)	Nucleotide Diversity (*π*) (SD)	Tajima’s *D*	Fu’s *Fs*
S7	Northern	Hanoi	Adult and Larvae	15	8	0.9048 (0.0502)	0.0112 (0.0062)	0.8382	0.9623
S8		Hai Phong	Adult and Larvae	15	7	0.7905 (0.1049)	0.0089 (0.0051)	−0.1512	0.3057
S9		Nam Dinh	Adult and Larvae	15	7	0.8286 (0.0823)	0.0101 (0.0056)	0.5880	1.6294
S12	Central	Da Nang	Adult and Larvae	15	9	0.8667 (0.0567)	0.0074 (0.0047)	−0.6752	0.6688
S14		Khanh Hoa	Adult and Larvae	15	10	0.9238 (0.0530)	0.0103 (0.0058)	0.4323	−1.1288
S15		Dak Lak	Adult and Larvae	15	7	0.8190 (0.0818)	0.0097 (0.0055)	0.9202	1.4991
S17	Southern	Ho Chi Minh city	Adult and Larvae	17	12	0.9412 (0.0432)	0.0104 (0.0058)	0.0465	−2.3716
S18		Can Tho	Adult and Larvae	15	8	0.8901 (0.0603)	0.0077 (0.0045)	−0.2074	−0.3867
**Total**				122	29	0.8730	0.0095	−0.1105	−1.0729

N, number of individuals analyzed.

## Data Availability

Haplotype sequences are available on GenBank, accession: OM883877-OM883905.

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
