# Peer review of "Invasion Pattern of Aedes aegypti in the Native Range of Ae. albopictus in Vietnam Revealed by Biogeographic and Population Genetic Analysis"

_insects, 2022, doi:10.3390/insects13121079_

Round 1
Reviewer 1 Report
The yellow fever mosquito, Aedes aegypti, introduced to Vietnam a century ago, is a serious arbovirus vector primarily responible to outbreaks of vector born-diseases as dengue and chikungunya fever. The indigenous mosquito species, Aedes albopiuctus is considered a secondary vector of both DENV and CHIKV. Vector control is predominant strategy for reducing the incidence of those vector-born diseases. Therefore, the knowledge of biology and ecology / biogeography and genetic diversity of the mosquito vectors (distribution in different country regions with the emphasis upon localization of Aedes aegypti and Aedes albopictus breeding places mainly in urban and periurban environment) is essential for efficient mosquito/vector control.
The authors of submitted paper focused on current distribution of the mosquito species in northern temperate (mountainous) and southern tropical climatic regions in Vietnam. They assessed the invasion pattern of Aedes aegypti through investigation of the distribution, relative abundance with its competitor Aedes albopictus .They found a high level of genetic diversity of Aedes aegypti with two major genetic lineages.
The submitted study has significant potential for optimizing vector control strategies not only in Vietnam, but in all southeast Asia mosquito regions.
I recommend the submitted article to be published in MDPI Insect.
Author Response
Comments to the Author
The yellow fever mosquito, Aedes aegypti, introduced to Vietnam a century ago, is a serious arbovirus vector primarily responible to outbreaks of vector born-diseases as dengue and chikungunya fever. The indigenous mosquito species, Aedes albopiuctus is considered a secondary vector of both DENV and CHIKV. Vector control is predominant strategy for reducing the incidence of those vector-born diseases. Therefore, the knowledge of biology and ecology / biogeography and genetic diversity of the mosquito vectors (distribution in different country regions with the emphasis upon localization of Aedes aegypti and Aedes albopictus breeding places mainly in urban and periurban environment) is essential for efficient mosquito/vector control.
The authors of submitted paper focused on current distribution of the mosquito species in northern temperate (mountainous) and southern tropical climatic regions in Vietnam. They assessed the invasion pattern of Aedes aegypti through investigation of the distribution, relative abundance with its competitor Aedes albopictus. They found a high level of genetic diversity of Aedes aegypti with two major genetic lineages.
The submitted study has significant potential for optimizing vector control strategies not only in Vietnam, but in all southeast Asia mosquito regions.
I recommend the submitted article to be published in MDPI Insect.
Response: It seems the manuscript meet the requirements of reviewer, so there is no change. Thank you very much for your positive comments and suggestions that the paper deserves to be published in Insects.
Reviewer 2 Report
Duong et al. have surveyed Ae. aegypti and albopictus across Vietnam, in which the former is an invasive species. Although the two species are often coexisting, they find striking geographical disparities in where the two species are found. They also use barcodes to examine the Ae. aegypti, finding that many different invasive lineages have colonized Vietnam, and that the invaders seem roughly homogenous across the country.
I enjoyed reading this paper and have a few comments. I have one major comment that needs to be addressed before I can recommend publication. There are several moderate comments, most of which should be addressed by the authors, and a larger number of minor comments, most of which are grammar and syntax stuff, which I hope the authors find helpful but they can ignore if they would rather.
Major comment:
My only real concern with this paper is that the question of sampling effort has not been accounted for. To give one example of many: the authors argue that aegypti is more abundant than albopictus, at least in the south. However, their data do not actually show this: aegypti is more abundant in domestic and peri-domestic containers, but albopictus is more abundant in natural sites. We cannot actually conclude which species is more abundant without knowing the relative ratios of the sites; i.e., perhaps there are 100X more natural sites than there are domestic and peri-domestic sites, in which case there are far more albopictus than aegypti at every single site.
I see two possible solutions to this problem, both of which would be fine by me:
First, if you did do transects or something like that, so that the NT numbers given in Table 2 are actually proportional to their abundance in nature, then you need to describe that protocol (I suspect you did not do this, but am happy to be wrong).
Second, I think it is more likely that you sampled containers haphazardly. I am fine with this from an experimental design perspective, as long as you are careful never to draw conclusions about the absolute number of mosquitoes in any of these categories. Areas of the text that need to be addressed include:
-208-211: You need to say that used tires, etc, were the “most frequently sampled” not “used”, since your data in Table 2 reflect your own sampling, not the actual usage by the mosquitoes.
-213-215: Again, this only reflects your sampling scheme, but implies that this is true of all mosquito larvae; I recommend deleting this sentence.
-317-319: I think you can say that your data shows that aegypti is a superior competitor for particular larval development sites, but you definitely can’t say it is a superior competitor overall, since albopictus outcompetes it everywhere for natural sites, and you don’t know which type of site is more important.
-409, 414: I don’t think that you can that the density of aegypti or albopictus is lower or higher in any particular region for the same reason.
-Figure 3: The y-axis here is not meaningful since raw numbers of larvae cannot be compared across habitats or sites. I recommend instead presenting a series of 6 bargraphs that show the relative proportion of aegypti and albopictus in southern domestic containers, northern domestic containers, southern peri-domestic containers, etc. If your sampling effort was uneven from site to site, then you probably want to calculate a mean of means, i.e., figure out the aegypti domestic container proportion for S18, S17, etc all separately, and then average those.
-Similarly, I think that your ANOVA needs to be redone; as is, you can’t have raw abundances as your response variable without some measure of sampling effort. The best solution, IMO, is to have your response variable be proportion(aegypti) at each site (and ideally include “site” as a random effect nested inside region). At this point, I’m not sure an ANOVA is still what you want (I suppose it would depend on the distribution of proportion(aegypti) in your data), perhaps a logit GLM?
-I’m not sure I found all of the instances, but hopefully this clarified what I mean and you can find the rest.
Moderate comments:
143: I cannot find how many biogeographic regions you used for the ANOVA analysis or what sites are in which. This needs to be explained—see my comment below on line 191.
179: Purified how?
191: Based on Table S2, it looks like you actually did two different AMOVA analyses, with two different sets of geographic regions. I think you need to clarify here that you did two parallel AMOVAs and explain why. You should also be clear about what geographic regions you used—either refer the reader to Table S2 or add this information to Table 3. Finally, I think you should briefly explain why you divided up the 8 sites into the particular 2 or 3 regions that you used for the AMOVA—is this based on geographical features? Biogeography of other species? Some sort of a posteriori analysis of your barcodes? (Also, I think in Arlequin you can have nested levels in your AMOVA. So you could do this in a single test that has level 1: North vs Central/South and then level 2: Central vs South).
230: I think you mean ANOVA here, no? Also, probably you mean to point to Table S1?
256-259: I am very confused by this passage. You talk about how your D and Fs results are significant, but you do not provide any information about how you calculated significant values for these statistics or how to interpret them, nor do you ever present the p-values themselves. I’m not familiar with D/Fs significance testing, is this something implemented in Arlequin?
271: Which of the two AMOVAs are you talking about there? Also, you may want to remind the reader what your various phi’s correspond to.
474: This isn't actually true (the raw data is not found anywhere in the MS or supplement).
Minor comments:
17: should be “of Aedes aegypti”
19: note sure what “breeding types” means here. Do you mean “larval habitat” or something like that?
36: better “This genetic diversity is unlikely to be relevant to the …”
47: better “the progression of urbanization created …”
49: better “facilitated their current”
51: I don’t think there is any value in abbreviating dengue, chikungunya, etc. Why not just spell them out?
55: better “as shown in the case of Ae. albopictus and Ae. aegypti,”
59-60: please remove the double brackets
83-84: I’m not sure what this sentence means.
109: I recommend showing the pass (or at least latitude lines) on Figure 1, like many of your readers I am not quite sure where in Vietnam the 16th parallel lies.
134: better “were investigated, all of which were at least 500 m from the others.”
183: I recommend putting the GenBank numbers in the Data Availability Statement. When you put them here in the methods it makes it seem like you got them from somewhere else.
197: If you reference this table before any others, I would make it Table S1.
225: better “sites S1-S6”
322: better “contained larvae of both”
351: I think that it would be better to put this in the past tense, along the lines of “Indeed, the present study shows that habitat-dependent displacement of Ae. albopictus by Ae. aegypti has occurred in Vietnam.” Obviously the displacement happened at some point, since aegypti isn’t native, but we don’t really have any evidence from this paper that it is ongoing, since you provide only a single timepoint.
362-364: I found this clause confusing, maybe “agree with the hypothesis of so-and-so that aegypti has pre-adapted behavior traits that prime it to be a successful invader: 1, 2, 3, etc” or something similar.
369: I am skeptical that not invading a region in the first place can really be considered a strategy, especially since lack-of-invasion probably doesn’t come from lack of dispersal, but more likely from dispersed individuals failing to reproduce (which is really the worst possible strategy).
388: I don’t think that hypothesis two would really explain barcode COI variability, though, would it? More explanation is necessary, at least.
415: better “(sites S7-S11 on Figure 1; Table S5)”
416: “tropical regions are favorable to Ae. albopictus development”: reference?
Author Response
Comments and Suggestions for Authors
Duong et al. have surveyed Ae. aegypti and albopictus across Vietnam, in which the former is an invasive species. Although the two species are often coexisting, they find striking geographical disparities in where the two species are found. They also use barcodes to examine the Ae. aegypti, finding that many different invasive lineages have colonized Vietnam, and that the invaders seem roughly homogenous across the country.
I enjoyed reading this paper and have a few comments. I have one major comment that needs to be addressed before I can recommend publication. There are several moderate comments, most of which should be addressed by the authors, and a larger number of minor comments, most of which are grammar and syntax stuff, which I hope the authors find helpful but they can ignore if they would rather.
Major comments:
My only real concern with this paper is that the question of sampling effort has not been accounted for. To give one example of many: the authors argue that aegypti is more abundant than albopictus, at least in the south. However, their data do not actually show this: aegypti is more abundant in domestic and peri-domestic containers, but albopictus is more abundant in natural sites. We cannot actually conclude which species is more abundant without knowing the relative ratios of the sites; i.e., perhaps there are 100X more natural sites than there are domestic and peri-domestic sites, in which case there are far more albopictus than aegypti at every single site.
I see two possible solutions to this problem, both of which would be fine by me:
First, if you did do transects or something like that, so that the NT numbers given in Table 2 are actually proportional to their abundance in nature, then you need to describe that protocol (I suspect you did not do this, but am happy to be wrong).
Second, I think it is more likely that you sampled containers haphazardly. I am fine with this from an experimental design perspective, as long as you are careful never to draw conclusions about the absolute number of mosquitoes in any of these categories.
Response: Thank you very much for pointing this out. We apologize for this inconvenience caused by not carefully describe the sampling method. We edited the collection method and added a sampling design in the Supplementary File. As we investigated urban areas, it possibly assumes that man-made containers are dominant compared to natural containers. Therefore, the study’s results relatively reflect the actual proportional abundance of containers of Aedes in each study area. We also agree with your comment that it is impossible to investigate all the containers existing in a certain area since many inaccessible spaces even if we applied transect investigations. However, we hope that you can see our efforts in order to report the best scenario of Aedes vectors’ distribution and abundance in Vietnam.
The collection method has been edited as the following: “In each surveyed city, we chose a study site (one study site per city) with similar habitat conditions which included a variety of potential sampling points, such as resident houses, recreational parks, playgrounds, car repair stations, pagodas, and schools. At each study site (a circular area with 5 km in diameter, approximately 78.5 km2), we tried to establish 16 sampling points which were spaced equally in distance within the study site (16 sampling points x 18 study sites = 288 sampling points in total) (Figure S1). At each sampling point (area of each sampling point approximately 300 m2), all containers with water were checked and sampled. Although the designed sampling points were often blocked by obstacles such as buildings or ponds, we tried to maintain the distance > 500 m between the sampling points (Figure S1).”
(See page 4, lines 131-140)
Moderate comments
Comment 2: 143: I cannot find how many biogeographic regions you used for the ANOVA analysis or what sites are in which. This needs to be explained—see my comment below on line 191.
Response: Thank you for your suggestion. We described the biogeographic regions in Tables S2 but forgot to cite in the sentence. We added in the sentence as the following: “Meanwhile, genetic structure was estimated by hierarchical analysis of molecular variance (AMOVA) in ARLEQUIN, with 1000 permutations and populations grouped by geographic regions defined in Table 3 including (1) 3 groups (North, Centre and South) and (2) 2 groups (North and Centre + South) (Table S2). We assessed the statistical significance based on the degree of differentiation among regions (ΦCT), among populations within regions (ΦSC), and within populations (ΦST) using permutation tests with 1000 random permutations. (See pages 6-7, line 193-199)
Comment 3: 179: Purified how?
Response: we added information about purification in the main text as the following: “The validated PCR products were then purified enzymatically using Exonuclease I to degrade the residual PCR primers and Shrimp Alkaline Phosphatase to dephosphorylate the remaining dNTP (New England BioLabs, USA) and bidirectionally sequenced using an ABI Prism Sequencer 3130 (Applied Biosystems, Foster City, USA), and the resulting sequences of both reverse and forward directions were manually corrected using MEGA v.7 [48] to ensure correct alignment along codons. “. (See page 6, lines 179-185).
Comment 4: 191: Based on Table S2, it looks like you actually did two different AMOVA analyses, with two different sets of geographic regions. I think you need to clarify here that you did two parallel AMOVAs and explain why. You should also be clear about what geographic regions you used—either refer the reader to Table S2 or add this information to Table 3. Finally, I think you should briefly explain why you divided up the 8 sites into the particular 2 or 3 regions that you used for the AMOVA—is this based on geographical features? Biogeography of other species? Some sort of a posteriori analysis of your barcodes? (Also, I think in Arlequin you can have nested levels in your AMOVA. So you could do this in a single test that has level 1: North vs Central/South and then level 2: Central vs South).
Response: Thank you for these kind observations and suggestions. In this analysis, we separate the sites to different regions based on the geographical regions in Vietnam. We added one more column in the Table 3 to indicate regions and modified the method as the following:
“Meanwhile, genetic structure was estimated by hierarchical analysis of molecular variance (AMOVA) in ARLEQUIN, with 1000 permutations and populations grouped by geographic regions defined in Table 3 including (1) 3 groups (North, Centre and South) and (2) 2 groups (North and Centre + South). We assessed the statistical significance based on the degree of differentiation among regions (ΦCT), among populations within regions (ΦSC), and within populations (ΦST) using permutation tests with 1000 random permutations.” (See pages 6-7, lines 193-199)
Comment 5: 230: I think you mean ANOVA here, no? Also, probably you mean to point to Table S1?
Response: Thank you very much for this kind suggestion. The suggested correction has been made as the following: “Specifically, the presence and abundance of Ae. aegypti was negatively affected by the presence and abundance of Ae. albopictus in certain climatic regions and breeding sites (ANOVA test, P<0.00001; Figure 3 and Table S1).” (See pages 6-7, lines 236-239)
Comment 6: 256-259: I am very confused by this passage. You talk about how your D and Fs results are significant, but you do not provide any information about how you calculated significant values for these statistics or how to interpret them, nor do you ever present the p-values themselves. I’m not familiar with D/Fs significance testing, is this something implemented in Arlequin?
Response: We added more detail in the analysis of indices Tajima’s D and Fu’s Fs as the following:
“Tajima’s D and Fu’s Fs were estimated for each population using ARLEQUIN v. 3.5.2.2 in order to establish non-neutral evolution and deviation from mutation-drift equilibrium, with population expansion signatures inferred using significance values that were calculated by generating 1000 random samples.” (See page 6, lines 187-191)
Comment 7: 271: Which of the two AMOVAs are you talking about there? Also, you may want to remind the reader what your various phi’s correspond to.
Response: Thank you for figure it out. We clarified the paragraph as the following: “The AMOVA test when all populations were grouped into 3 regions indicated that most of the total genetic variance could be attributed to within-population variation, thereby suggesting very weak genetic structure and high gene flow among Ae. aegypti populations in Vietnam (ΦST = 99.36, P=0.43; Table S2). However, when all populations were divided into two geographic clusters (i.e., north and south), AMOVA analysis indicated significant spatial genetic structuring (ΦCT= 4.42, P<0.05); however, most of the variance still occurred within populations (ΦST = 97.88, P=0.43; Table S2).”. (See page 10, lines 277-283)
Comment 8: 474: This isn't actually true (the raw data is not found anywhere in the MS or supplement).
Response: The suggested correction has been made (see page 14, lines 482-483)
Minor comments
Comment 9: 17: should be “of Aedes aegypti”
Response: The suggested correction has been made. (See page 1, line 17)
Comment 10: 19: note sure what “breeding types” means here. Do you mean “larval habitat” or something like that?
Response: The suggested correction has been made. (See page 1, line 19)
Comment 11: 36: better “This genetic diversity is unlikely to be relevant to the …”
Response: The suggested correction has been made as the following: “This genetic diversity is unlikely to be relevant to the invasion success of Ae. aegypti.”. (See page 1, line 36-37)
Comment 12: 47: better “the progression of urbanization created …”
Response: The suggested correction has been made. (See page 2, lines 48-49)
Comment 13: 49: better “facilitated their current”
Response: The suggested correction has been made. (See page 2, line 50)
Comment 14: 51: I don’t think there is any value in abbreviating dengue, chikungunya, etc. Why not just spell them out?
Response: The suggested correction has been made. (See page 2, line 52)
Comment 15: 55: better “as shown in the case of Ae. albopictus and Ae. aegypti,”
Response: The suggested correction has been made. (See page 2, lines 56-57)
Comment 16: 59-60: please remove the double brackets
Response: The suggested correction has been made. (See page 2, line 60-61)
Comment 17: 83-84: I’m not sure what this sentence means.
Response: The sentence has been modified as the following: “After a century, the species has become a mosquito vector primarily responsible to outbreaks of vector-borne diseases, including dengue, chikungunya, and Zika [35–37].”. (See page 2, lines 84-86)
Comment 18: 109: I recommend showing the pass (or at least latitude lines) on Figure 1, like many of your readers I am not quite sure where in Vietnam the 16th parallel lies.
Response: Thank you for pointing this out. We think that the information is not really necessary, so we deleted the part and modified the sentence. (See page 3, lines 107-108)
Comment 19: 134: better “were investigated, all of which were at least 500 m from the others.”
Response: The suggested correction has been made. (See page 4, lines 139-140)
Comment 20: 183: I recommend putting the GenBank numbers in the Data Availability Statement. When you put them here in the methods it makes it seem like you got them from somewhere else.
Response: The suggested correction has been made as the following: “Data Availability Statement: haplotype sequences are available on GenBank, accession: OM883877-OM883905.”. (See page 14, line 482-483)
Comment 21: 197: If you reference this table before any others, I would make it Table S1.
Response: Thank you for your suggestion but we have mentioned Table S1 (page 4, line 144) and Table S2 (page 6, line 188) before Table S3 in line 201. Therefore, we will keep the Table S3 in this sentence (See page 7, line 205).
Comment 22: 225: better “sites S1-S6”
Response: The suggested correction has been made. (See page 7, line 233)
Comment 23: 322: better “contained larvae of both”
Response: The suggested correction has been made. (See page 11, line 332)
Comment 24: 351: I think that it would be better to put this in the past tense, along the lines of “Indeed, the present study shows that habitat-dependent displacement of Ae. albopictus by Ae. aegypti has occurred in Vietnam.” Obviously the displacement happened at some point, since aegypti isn’t native, but we don’t really have any evidence from this paper that it is ongoing, since you provide only a single time point.
Response: The suggested correction has been made. (See page 11, line 360-362)
Comment 25: 362-364: I found this clause confusing, maybe “agree with the hypothesis of so-and-so that aegypti has pre-adapted behavior traits that prime it to be a successful invader: 1, 2, 3, etc” or something similar.
Response: Thank you for your suggestion. The sentence has been modified as your suggestion: “ This study’s findings agree with the hypothesis of pre-adapt behavioral traits of domestic Ae. aegypti that prime it to be a successful invader: (1) house entry (endophily); (2) the preference for oviposition in human-made domestic containers; (3) preference for blood-feeding from humans [20].” (See page 12, line 372-375)
Comment 26: 369: I am skeptical that not invading a region in the first place can really be considered a strategy, especially since lack-of-invasion probably doesn’t come from lack of dispersal, but more likely from dispersed individuals failing to reproduce (which is really the worst possible strategy).
Response: We agree with your comment that successful reproduction is important. In this sentence, we aim to describe the invasion strategy of Ae. aegypti in northern Vietnam, in which Ae. aegypti was more found in the houses rather than outside. It could help them prevent their closest competitor, Ae. albopictus, closer to human and warmer than outside. Due to your comment, we modified the sentence as the following: “This behavior can be considered as a hidden strategy that results in reduced interspecific competition in regard to mating interference, blood sources, larval competition, or other factors (e.g. blood sources, temperature).” (See page 12, lines 378-380).
Comment 27: 388: I don’t think that hypothesis two would really explain barcode COI variability, though, would it? More explanation is necessary, at least.
Response: Thank you for your comments and suggestion. Actually, the hypothesis 2 is possible in nature but requires a long period of time in terms of using COI gene. The sentence has been deleted (See page 12, lines 396-397)
Comment 28: 415: better “(sites S7-S11 on Figure 1; Table S5)”
Response: The sentence has been modified as the reviewer’s suggestion (See page 13, line 423)
Comment 29: 416: “tropical regions are favorable to Ae. albopictus development”: reference?
Response: We added a reference to this opinion already (See page 13, line 425)
Reviewer 3 Report
The manuscript “Invasion pattern of Aedes aegypti in the native range of Ae. albopictus in Vietnam revealed by biogeographic and population genetic analysis” by Duong et al. studied data of absolute numbers of immature Aedes specimens collected in 18 sites throughout Vietnam. Collections were performed in three different habitats: domestic, peri-domestic, and natural. A subset of Ae. aegypti collected was used for COI sequencing, then used for phylogenetics and population genetics analysis.
I have three points that I would like further comments on:
- It is not clear to me if the efforts to collect in the three habitats were similar in every site. For instance, is the difference in Aedes species abundance/presence due to geographic distribution or in the north the work to collect in natural habitat was more intense? Also, were tires the most frequent breeding site used by Aedes or were they the most surveyed type of breeding site?
- I missed a discussion about the level of urbanization. Is there a difference between north and south of Vietnam?
- Conclusion about Ae. aegypti populations being associated with invasive populations from other parts of the world is weak because sequenced from Africa is far underrepresented in the analysis.
Author Response
Comments and Suggestions for Authors
The manuscript “Invasion pattern of Aedes aegypti in the native range of Ae. albopictus in Vietnam revealed by biogeographic and population genetic analysis” by Duong et al. studied data of absolute numbers of immature Aedes specimens collected in 18 sites throughout Vietnam. Collections were performed in three different habitats: domestic, peri-domestic, and natural. A subset of Ae. aegypti collected was used for COI sequencing, then used for phylogenetics and population genetics analysis.
Comment 1: It is not clear to me if the efforts to collect in the three habitats were similar in every site. For instance, is the difference in Aedes species abundance/presence due to geographic distribution or in the north the work to collect in natural habitat was more intense? Also, were tires the most frequent breeding site used by Aedes or were they the most surveyed type of breeding site?
Response: Thank you very much for pointing this out. We apologize for this inconvenience caused by not carefully describe the sampling method. We edited the collection method and added a sampling design in the Supplementary File. As we investigated urban areas, it possibly assumes that man-made containers are dominant compared to natural containers. Therefore, the study’s results relatively reflect the actual proportional abundance of containers of Aedes in each study area. We also agree with your comment that it is impossible to investigate all the containers existing in a certain area since many inaccessible spaces even if we applied transect investigations. However, we hope that you can see our efforts in order to report the best scenario of Aedes vectors’ distribution and abundance in Vietnam.
The collection method has been edited as the following: “In each surveyed city, we chose a study site (one study site per city) with similar habitat conditions which included a variety of potential sampling points, such as resident houses, recreational parks, playgrounds, car repair stations, pagodas, and schools. At each study site (a circular area with 5 km in diameter, approximately 78.5 km2), we tried to establish 16 sampling points which were spaced equally in distance within the study site (16 sampling points x 18 study sites = 288 sampling points in total) (Figure S1). At each sampling point (area of each sampling point approximately 300 m2), all containers with water were checked and sampled. Although the designed sampling points were often blocked by obstacles such as buildings or ponds, we tried to maintain the distance > 500 m between the sampling points (Figure S1).”
(See page 4, lines 131-140)
Comment 2: I missed a discussion about the level of urbanization. Is there a difference between north and south of Vietnam?
Response: The level of urbanization in Vietnam differ intensively between cities, especially, Hanoi and Ho Chi Minh city compared to mountainous cities (S1-Lai Chau, S2-Dien Bien, S15-Gia Lai). However, there is no difference comparing northern and southern cities.
Comment 3: Conclusion about Ae. aegypti populations being associated with invasive populations from other parts of the world is weak because sequenced from Africa is far underrepresented in the analysis.
Response: In the phylogenetic tree, we used 9 representative sequences from Africa ranked second comparing to other regions. However, as your suggestion, we cannot surely assume that the source from other invasive regions, therefore, we have modified the sentences as the following: “The sources of Ae. aegypti in Vietnam is likely genetically similar to other invasive ranges worldwide. Further, we found a high level of genetic diversity through the country.” (See page 13, lines 463-465).
Round 2
Reviewer 2 Report
The revisions look great! I have only a couple minor points:
The added information about the sampling design is fantastic and addresses most of my concerns. I do think it might be worth noting in the discussion that because your sampling was of urban areas, your finding that aegypti is competitively dominant in the south may not necessarily apply to southern rural areas; I can imagine that the greater host-generality of albo gives it an advantage when the density of humans is low. Still, from a public health perspective the urban perspective that you give is probably the most important.
Regarding lines 264-265, I think that you should say that "Neutrality tests (Tajima's D and Fu's Fs) did not yield significant results (all p > 0.05), whether for the population as a whole or each local population individually, which suggested that the observed ...." Looking at the Arlequin manual it seems that the null hypothesis is neutrality, so the "significant" result would be non-neutrality if the p < 0.05 (or 0.02 in the case of Fu's Fs: http://cmpg.unibe.ch/software/arlequin35/man/Arlequin35.pdf 8.1.7.4-5).
Author Response
Responses to the comments of Reviewer 2
Comments from Reviewer 2
Comments and Suggestions for Authors
The added information about the sampling design is fantastic and addresses most of my concerns. I do think it might be worth noting in the discussion that because your sampling was of urban areas, your finding that aegypti is competitively dominant in the south may not necessarily apply to southern rural areas; I can imagine that the greater host-generality of albo gives it an advantage when the density of humans is low. Still, from a public health perspective the urban perspective that you give is probably the most important.
Response: Thank you very much for pointing this out. We added this information as the following: “Furthermore, because this study was conducted in urban areas across Vietnam, investigations must be undertaken in other areas, such as rural or forest areas, to examine the invasion pattern of Ae. aegypti, especially in the southern ranges.” (See page 12, lines 394-396)
Regarding lines 264-265, I think that you should say that "Neutrality tests (Tajima's D and Fu's Fs) did not yield significant results (all p > 0.05), whether for the population as a whole or each local population individually, which suggested that the observed ...." Looking at the Arlequin manual it seems that the null hypothesis is neutrality, so the "significant" result would be non-neutrality if the p < 0.05 (or 0.02 in the case of Fu's Fs: http://cmpg.unibe.ch/software/arlequin35/man/Arlequin35.pdf 8.1.7.4-5).
Response: We apologize for this errors, it is actually insignificant results. We edited in the paragraph: “Neutrality tests (Tajima’s D and Fu’s Fs) yielded insignificant results (p > 0.05) for the population as a whole or each local population individually, indicating that the observed polymorphisms were consistent with the neutral mutation model (Table 3, Figure 4).” (See page 8, lines 266-269).